# Multi-Micronutrient Fortified Rice Improved Serum Zinc and Folate Concentrations of Cambodian School Children. A Double-Blinded Cluster-Randomized Controlled Trial

**DOI:** 10.3390/nu11122843

**Published:** 2019-11-20

**Authors:** Khov Kuong, Pety Tor, Marlene Perignon, Marion Fiorentino, Chhoun Chamnan, Jacques Berger, Kurt Burja, Marjoleine A. Dijkhuizen, Megan Parker, Nanna Roos, Frank T. Wieringa

**Affiliations:** 1Department of Fisheries Post-Harvest Technologies and Quality control (DFPTQ), Fisheries Administration, Ministry of Agriculture, Forestry and Fisheries, Phnom Penh, Cambodia; chhounchamnan@gmail.com; 2Department of Nutrition, Exercise and Sports (NEXS), University of Copenhagen, Rolighedsvej 26, 1958 Frederiksberg, Denmark; madijkhuizen@gmail.com (M.A.D.); nro@nexs.ku.dk (N.R.); 3Institut Pasteur du Cambodge, Phnom Penh, Cambodia; 4UMR-204 Nutripass, Institut de Recherche pour le Développement (IRD), IRD/Université de Montpellier/SupAgro, 34394 Montpellier, France; marlene.perignon@inra.fr (M.P.); marionfiorentino@hotmail.com (M.F.); jacques.berger@ird.fr (J.B.); 5World Food Programme (WFP), Phnom Penh, Cambodia; kurt.burja@wfp.org; 6PATH (Program for Appropriate Technology in Health), 2201 Westlake Avenue, Suite 200, Seattle, WA 98121, USA; mparker@path.org

**Keywords:** fortified rice, micronutrient, deficiency, zinc, school children, Cambodia

## Abstract

Background: Within Cambodia, micronutrient deficiencies continue to be prevalent in vulnerable groups, such as women and children. Fortification of staple foods such as rice could be a promising strategy for Cambodia to improve micronutrient status. Objective: Our objective was to investigate the impact of multiple-micronutrient fortified rice (MMFR), distributed through a World Food Program school-meals program (WFP-SMP) on serum zinc concentrations and folate status in a double-blind, cluster-randomized, placebo-controlled trial. Methods: Sixteen schools were randomly assigned to receive one of three different types of extruded-fortified rice (UltraRice Original (URO), UltraRice New (URN), or NutriRice) or unfortified rice (placebo) six days a week for six months. A total of 1950 schoolchildren (6–16 years old) participated in the study. Serum zinc (all groups) and folate (only in NutriRice and placebo group) concentrations were assessed from morning non-fasting antecubital blood samples and were measured at three time points (baseline and after three and six months). Results: After six months of intervention, serum zinc concentrations were significantly increased in all fortified rice group compared to placebo and baseline (0.98, 0.85 and 1.40 µmol/L for URO, URN and NutriRice, respectively) (interaction effect: *p* < 0.001 for all). Children in the intervention groups had a risk of zinc deficiencies of around one third (0.35, 039, and 0.28 for URO, URN, and NutriRice, respectively) compared to the placebo (*p* < 0.001 for all). The children receiving NutriRice had higher serum folate concentrations at endline compared to children receiving normal rice (+2.25 ng/mL, *p* = 0.007). Conclusions: This study showed that the high prevalence of zinc and folate deficiency in Cambodia can be improved through the provision of MMFR. As rice is the staple diet for Cambodia, MMFR should be considered to be included in the school meal program and possibilities should be explored to introduce MMFR to the general population.

## 1. Introduction

Micronutrient deficiencies remain a critical public health issue in most developing countries, affecting about one third of children and women in reproductive age [1,2]. Deficiencies in important micronutrients including vitamin A, iron, zinc, and folate contribute to poor growth, impaired cognitive development, and increase the risk of morbidity and mortality from infectious diseases [3]. This can lead to negative impacts on socio-economic development at individual, community, and national levels [4]. For example, in Cambodia, the annual costs of stunting and deficiencies of iron, iodine, vitamin A, and zinc was estimated at 266 million US$ [5].Combating micronutrient deficiencies among vulnerable groups has been a public health priority for both developing and developed countries. The Copenhagen Consensus 2012 has put combating nutritional deficiencies as one of the most effective strategies towards improving health [6], especially in low- and middle-income countries, where the burden of micronutrient malnutrition is enormous. 

In Cambodia, despite a considerable reduction in national poverty since the mid-1990s, under-nutrition remains a major problem. In 2014, stunting and anemia affected 32.4% and 43.6% of children under 5 years, respectively [7]. Micronutrient deficiencies among preschool children was also highly prevalent, with 67.5% and 20.2% of them being classified as deficient in zinc (plasma concentrations <9.9 μmol/L) and folate (plasma concentration <10 nmol/L), respectively [8]. While the prevalence of frank iron deficiency was low (<10%), 24.7% of preschool children had marginal iron stores [8].

Deficiency in zinc is associated with suboptimal linear growth and with impaired immune function, leading to recurrent infections [9]. Supplementation of zinc was shown to improve height and weight [10] and reduce the incidence of pneumonia and acute diarrhea and in certain groups mortality [11,12]. There is some evidence from observational and intervention studies in schoolchildren that zinc may benefit cognitive performance [13], though the evidences are limited and inconsistent. Folate plays a key role in nucleic acid synthesis and deficiency leads to megaloblastic macrocytic anemia [14]. Folate has been associated with brain development in children which begins prenatally and continues through school age [15]. Moreover, folate deficiency in pregnant women contributes to neural tube defects and prematurity of birth [16].

Micronutrient deficiencies and malnutrition are widespread among schoolchildren. A review from 2010 estimated that zinc, iron, and vitamin A deficiencies affect 20–30% of school-age children in South-East Asia [17]. These deficiencies can impair physical and mental development during the important school years and reduce school attendance by increasing the likelihood of common morbidities. Some studies have reported that it is still possible to improve cognition during the primary school years by improving micronutrient status [18,19], and reducing morbidity incidence and growth deficits; however, the overall effects of these factors on cognition were equivocal and more evidence is required from studies in different contexts.

Fortification of staple foods such as rice could be a promising strategy for Cambodia, where approximately 70% of the daily energy intake comes from rice [20]. The fortification of staple foods is advantageous because it does not require the target population to change their dietary habits and allows fortification with multiple micronutrients since deficiencies often co-occur [21]. However, the Cambodian government needs evidence on the impact of fortified rice on nutrition, health, and development in school-age children to support the inclusion of fortified rice in country-wide programs, as well as to support future national food fortification guidelines.

The Fortified Rice for Schoolchildren in CAmbodia (FORISCA) UltraRice + NutriRice project was a cluster-randomized, double-blinded, placebo-controlled trial, conducted in 16 primary schools, all situated in one province in Cambodia. The project aimed to quantify the impact of multi-micronutrient fortified rice (MMFR), which was distributed through the World Food Program (WFP) school meal program as a single meal per day, on micronutrient status, health and cognition of Cambodian schoolchildren. The current paper investigates the impact of MMFR on serum zinc and folate status in schoolchildren. Other outcomes of FORISCA have been reported elsewhere [22].

## 2. Materials and Methods 

### 2.1. Study Design and Population

In a double-blinded, cluster-randomized, placebo-controlled trial, three different types of multi-micronutrient fortified rice were introduced through the WFP school meal program in Cambodia (22). Schools served as cluster, with in total 16 primary schools in rural Kampong Speu province being randomized to one of four interventions (three extruded fortified rice groups and one placebo rice group). The three extruded rice groups received either UltraRice original formula (URO), UltraRice new formula (URN) or NutriRice produced by company DSM/Buhler. Schools were eligible if they participated in the WFP school meal program and all children were served breakfast daily. In total, 18 schools were eligible, of which two were excluded because one school had double the number of school children (*N* > 1200) as the other schools, and one school had less than 100 children; for the biochemical determination of micronutrient status, a minimum of 125 school children was required per school. The sample size was calculated based on the intention to detect changes in anemia prevalence, hemoglobin concentration and a number of micronutrient statuses. The sample size of 500 children per intervention group was decided on the ability to detect a difference in hemoglobin (Hb) concentration between placebo and intervention groups of at least 3 g/L, with a power of 0.90 and statistical significance of 0.05, assuming an average Hb concentration of 110 g/L and an SD of 15 g/L. All other outcomes needed smaller sample sizes. Due to limited funding, only one type of fortified rice (NutriRice) was selected to study the effect of MMFR on folate status as compared to placebo. The trial took place from November 2012 to June 2013. Written informed consent of at least one parent was obtained prior to the study. Ethical approval was obtained from the Cambodian Ministry of Health, Education and Planning and the Ethical Review board of PATH, USA. This study population is described in full details by Perignon et al. [22].

### 2.2. Intervention 

The three types of fortified rice differed in micronutrient compositions (Table 1), as well as production methods. URO and URN was produced by UltraRice Technology from PATH [23]. URO did not contain vitamin A but had a higher amount of iron. The fortified kernels were produced with a technique that used lower temperatures than the two other fortified kernels (“cold extrusion”). URN was the improved fortified kernel which contained vitamin A and other B vitamin produced with hot extrusion. NutriRice was produced with hot extrusion also, by DSM/Buhler. The school meals consisted of rice, canned fish, vitamin A + D fortified vegetable oil, yellow split peas, and iodized salt. After baseline data collection, all children were dewormed by a single dose of 500 mg mebendazole. 

### 2.3. Food Preparation

Preparation and cooking of rice was conducted in the school kitchens using the same traditional method applied to all participating schools. Excess amount of water was added to the rice before cooking and heated until it boiled, which took approximately 15 min. The excessive water was then removed. Cooking was continued for another 10 min until all water was completely absorbed. Canned fish were cooked separately for a soup to which oil and salt were added. For preparation of peas, some schools boiled them together with the rice while other schools boiled separately and mixed before serving. The cooked food was distributed in buckets to each class. 

### 2.4. Randomization and Blinding

From the 16 schools eligible, four schools were randomly assigned to each of the intervention groups using a four-layered stratification of school size to ensure that the intervention groups has equal representation of different school sizes [22]. The randomization was done using www.randomization.com to obtain a final group sizes within 10% difference of each other. Each intervention group was randomly selected to receive either URO, URN, NutriRice, or normal rice (Placebo). To strengthen the blinding, each intervention arm of four schools was split into two groups of two schools, each given a letter code (A–H). The codes were placed in a sealed envelope and kept safe at WFP office in Phnom Penh to be opened only after primary analyses had been completed. Randomization and assignment of codes was conducted by one researcher (MAD) who was not involved in the field work and the codes were not known by researchers or field staffs during implementation assuring double-blinded trial. Blending of rice with fortified kernels was done under supervision of WFP at a local food factory in Phnom Penh. Rice was packaged in bags labeled with code according to allocation to intervention group. 

For participating children of each school, 132 children per school were randomly selected, stratified for gender and grade. This was 10% more than required sample (125 per school), as it was expected that some children were not available at recruitment day, e.g., due to illness or refusing to participate. 

### 2.5. Blood Sample and Measurement 

Non-fasting blood samples were taken from the antecubital vein in the morning between 8:00 AM to 12:00 PM using venipuncture following the standard protocol. Blood (5 mL) was stored in trace-element free vacutainers with no anticoagulant (Vacuette, Greiner Bio One, Austria). Blood samples were stored in cool-boxes kept cool with plastic ice packs at temperature of <5 °C and transported to Phnom Penh within 5 h of collection. At the laboratory facility in Phnom Penh, blood samples were centrifuged at 2700 rpm (1300 g) for 10 min at room temperature. 

Serum samples were then aliquoted in capped Eppendorf tubes and stored at −25 °C until transfer for analysis. Serum samples were sent on dry ice to National Institute of Nutrition (Hanoi, Vietnam) for zinc analysis. Zinc concentration was measured using a flame atomic absorption spectrophotometer (GBC, Avanta +) using trace element-free procedures. Serum from a sub-sample of the study population was sent to the Institut Pasteur du Cambodge (Phnom Penh, Cambodia) for analysis of folate concentration by automated radioimmunoassay (Cobas e411 analyzer, Roche, Japan). Zinc deficiency was defined using the following cut-offs: serum zinc <9.9 µmol/L for all children aged 4–9 years, <10.1 µmol/L for girls aged 10 yrs and up, <10.7 µmol/L for boys aged 10 yrs and up [24]. Severe zinc deficiency was defined as serum zinc <7.6 μmol/L. Folate deficiency was defined by serum folate concentration <4 ng/ml (10 nmol/L) [25]. Measurement of other biomarkers are described by Perignon et al. [22].

### 2.6. School Attendance

School attendance was recorded every school day. Two school monitors were hired for each school to monitor attendance of each class. They visited the child’s home when the child was absent to record the reason for absence. School attendance was calculated from the attendance of all days the school operated starting from delivery of fortified rice after baseline data collection until the day of endline data collection. The fortified rice for the study was delivered at 14th December 2012 to all schools. Data collection for ML and EL was conducted over a period of 3 weeks in February–March and June–July, respectively. In each intervention group, each child received a school meal for 126 to 132 days. There were more school days in the BL–ML period compared to ML–EL period due to school vacation of 2 weeks and number of national holiday during the ML–EL period.

### 2.7. Estimation of Zinc Intake

Zinc intake contributed from the intervention meal was estimated based on an average rice intake of 115 g per meal for each child. The estimated total zinc intake from the intervention foods per child over the study period was calculated by multiplication of the daily zinc intake and the number of days receiving fortified rice or placebo rice.

### 2.8. Data Management and Statistical Analysis

Data were entered in Epidata version 3.1 by two independent data entry operators. Data quality was ensured by matching the two sets of entries and verifying records. Statistical analyses were carried out using SPSS software version 22.0 (IBM corp). Only children whose data on zinc concentrations were available at baseline, midline, and endline were included in the final analysis. For folate concentrations, only data on baseline and endline were available. Normality was evaluated using the Kolmogorov-Smirnoff test. Continuous variables with skewness and kurtosis values outside the range of −1.0 to +1.0 were log-transformed. The analysis to quantify the impact of the intervention took into account the random effects of individuals and school clusters, using generalized mixed models (linear or binary logistic regression). Generalized mixed models were performed to evaluate the effects of time, group and time × group interaction on zinc status and prevalence of zinc deficiency with adjustment for age, gender and baseline characteristics including hemoglobinopathy, body iron, haemoglobin concentration, anemia, parasite infection, and inflammation. Multiple comparisons were conducted by using the Bonferroni post-hoc test.

## 3. Results

Rice kernels were analyzed for micronutrient composition by Silliker (Markham, Ontario, Canada) (for both UltraRice kernels) and Buhler (NutriRice kernels; Table 1). A flow chart of all study participants is shown in Figure 1. At baseline, two children were excluded due to severe anemia (and received multiple micronutrient supplements for two months as per protocol). In total, 1960 children were eligible children at recruitment. Zinc data were available for 1858 children at baseline. During midline and endline follow up, data were available for 1524 (82% of BL) and 1667 (90% of BL) children, respectively.

### 3.1. Baseline Characteristics 

Baseline characteristics of the participants in each group are presented in Table 2. The prevalence of stunting was 43.2%, of which 13.6% was classified as severe stunting. Almost all (89.7%) of the children were classified as zinc deficient and approximately half (48.9%) were severely zinc deficient. Folate status was only available for placebo and NutriRice group. Prevalence of folate deficiencies were 18.0% and 9.1% in placebo and NutriRice group, respectively. 

### 3.2. School Attendance and Dietary Intake

Table 3 shows the days of school attendance as an indicator for the number of meals, and the nutritional intake of zinc and folic acid in the intervention groups. The total school attendance for the periods between sampling points, as well as immediately (10 days) before blood sampling, was similar among the groups (82–90%). Zinc intake per each group for each child ranged from 2.3 to 4.2 mg per meal (day), which contributed about 29–53% of the Recommended Daily Allowance (RDA) for zinc (of 8 mg/d). Phytate content in cooked rice was low (109 mg/100 g) and phytate: zinc molar ratio was 3.55 in the URO group, 5.34 in the URN group, and 2.93 in the NutriRice group. The intake of folic acid per meal (day) was highest in URN (0.32 mg) compared to NutriRice (0.16 mg) and URO (0.2 mg). 

### 3.3. Effect of Intervention on Zinc Concentration

Serum zinc concentrations at endline were significantly higher in all the intervention groups compared with the placebo group (interaction effect: *p* < 0.001 for all) (Table 4). The serum zinc concentrations in the URO, URN, and NutriRice groups were 0.98, 0.85, and 1.40 µmol/L higher, respectively, compared to placebo. Moreover, at endline, i.e., after six months of intervention, serum zinc concentrations were significantly increased in all fortified rice group compared to baseline. During midline evaluation after three months of intervention, the serum zinc concentration was elevated compared to baseline in all intervention groups as well as placebo. At midline, zinc concentrations in the URO and URN groups were significantly lower than placebo with −1.02 and −0.50 µmol/L for URO and URN respectively (*p* < 0.05 for both). 

### 3.4. Effect of Intervention on Prevalence of Zinc Deficiency

At endline, after six months of intervention, the prevalence of low serum zinc concentrations and severe zinc deficiency were significantly lower in all intervention groups, compared to placebo (Table 5). During midline, after 3 months of the intervention, children in the URO group had a higher prevalence of zinc deficiency (*p* = 0.008), but not severe zinc deficiency, in comparison to the placebo group. After six months, the children in the URO and URN groups had a risk of one fourth (Odds Ration (OR) = 0.25, *p* < 0.001) and NutriRice group a risk of one sixth (OR = 0.16, *p* < 0.001) as compared to placebo group for being zinc deficient. Children in the intervention groups had a risk of around one third (0.35, 039, and 0.28 for URO, URN, and NutriRice, respectively), compared to the placebo group for having severe zinc deficiency (*p* < 0.001 for all). 

### 3.5. Effect of Intervention on Folate Status

Serum folate concentrations increased over the 6 months intervention period in both the placebo and NutriRice groups (Table 6), but the increase in the NutriRice group was significantly larger than in the placebo group (*p* = 0.007). The children receiving NutriRice significantly increased their serum folate concentration by 2.25 ng/mL compared to placebo over the period of 6-month intervention (*p* = 0.007). 

In both groups, the prevalence of folate deficiency was reduced, from 18% to 2%, and 9.1% to 0% for the placebo and NutriRice groups, respectively. However, the change in deficiency prevalence was not significant for both groups at endline in comparison to baseline.

## 4. Discussion

### 4.1. MMFR Increased Zinc Concentration and Reduced Deficiency

This study showed that the provision of MMFR in a school meal program had a significant and positive impact on serum zinc concentrations and lowered the prevalence of zinc deficiency among the school children in comparison to an unfortified rice (placebo) group. The largest reduction in the prevalence of zinc deficiency was seen in the intervention group with the highest zinc content in the fortified rice. Similar findings of fortified rice were reported from a study in school children in Thailand [26], in which rice fortified with zinc, iron, and vitamin A provided over a five-month period increased the serum zinc concentration by 1.9 µmol/L, from 9.4 µmol/L at baseline to 11.3 µmol/L at endpoint. The larger impact documented in the Thai study can be explained by the higher level of zinc fortification, which was 9 mg/day, compared to an average additional 2.3–4.2 mg of zinc per day in the present study. 

In this study, serum zinc concentration was used as an indicator for zinc status, following the evidence of a strong relationship between dietary zinc intake and serum zinc concentration as reported by Hess et al. [27]. When zinc intake was increased over a short period (1–2 weeks), serum zinc concentrations rose sharply with additional zinc provided, reaching a plateau around the intake level of 25–30 mg/d. This short-term reflection of dietary intake on serum zinc explained the finding that serum zinc concentrations did not rise continuously over the course of the intervention period. At the midline point, after 3 months, serum zinc concentrations were raised in all groups, including the placebo group. This rise of zinc concentrations is likely to reflect a seasonal rise in dietary zinc from food sources other than the fortified rice. The period from baseline to midline (December-March) is the peak harvest season in Cambodia for rice, fruits, and particularly the abundance of cheap fish. About 40% of catches during peak season from December to March are the small fish captured from Bag-net river fisheries [28], with prices ranging from 0.3–0.5 USS per kg [29]. Fish consumption varied from 215 g to 525 g/household/day from low to high catch season [30]. Roos et al. estimated that small fish made up 50–80% of all fish eaten during the fish harvest season in Cambodia and they are eaten whole and therefore are a rich source in zinc and other micronutrients [31]. Therefore, the increased access to this zinc-dense food is suggested to be reflected in the sharp increase of serum zinc concentration at midline point. There was no seasonal data on dietary patterns in the study area. Between midline and endline, the intervention spanned the lean period (April–July), children had less access to nutritious foods outside of the school meal, and hence, the fortified rice played a more important role to fill the gap of zinc intake during that period. Overall, zinc fortification of rice could help raise zinc intakes in the zinc deficient population, thereby improving zinc status throughout the year, regardless of seasonal changes in dietary intakes. 

Previous studies in young and school children reported an increase in serum zinc concentration that resulted from zinc supplementation but not from zinc fortified foods, although zinc content was the same or slightly higher in fortified foods, suggesting that zinc was maybe less well absorbed due to limited bioavailability of zinc in those food vehicles [32,33,34,35]. Studies in men and young children in Senegal [36,37] reported that serum zinc responded within 2 weeks in subjects receiving supplements but not in subjects consuming zinc-fortified foods. Another reason for the failure to show the efficacy of zinc-fortified foods include relatively high serum zinc concentrations at baseline and a food vehicle inhibitory to zinc absorption [38,39]. Our success in demonstrating an increase in serum zinc by incorporating the multiple micronutrient fortified rice into the school meal program could be due to the low phytate content of the fortification vehicle, and the low zinc status of the study children at baseline. Several studies have been conducted to measure zinc absorption from high- or low-phytate foods that have been fortified with zinc and showed that phytate:zinc molar ratio of the meals was the major factor accounting for the observed differences in fractional absorption of zinc [39,40]. Phytate content in our fortified rice was very low (108.9 mg/100 g) providing the molar ratio of phytate:zinc ranged from 2.93 to 5.34 across the different intervention group. The phytate:zinc molar ratio of fortified rice in the present study were much lower compared to other fortified food vehicles including wheat, maize, and other cereal [41], as well as lower than the standard cut-off suggested by the World Health Organization (WHO) which is a ratio below 15. This may explain why studies on zinc added to wheat and maize products did not improve zinc status [32,34,42].

Aaron et al. [36] also reported that serum zinc responded quickly to short term supplementation but not to food fortification, suggesting that serum zinc may not be as useful for monitoring short-term fortification programs. However, in our study, the intake of zinc from fortified rice was higher in baseline to midline period (BL-ML) than in midline to endline (ML-EL) period due to school vacation and many national holidays during the later. Intake within BL-ML compared to ML-EL could also explain the higher serum concentration at ML compared to EL and this suggests that serum zinc levels are responsive to consumption of fortified foods within a 3 month period, contrary to the findings of Aaron et al. In the present study, the school attendance of children for 10 days before data collection at ML and EL did not differ between rice groups thus this cannot be considered to have contributed to the difference in zinc concentrations. Response of increased serum zinc to a consumption of short (or maybe medium) term of zinc-fortified food as in our study and studies in Thailand [26] suggested that zinc fortified food need to be provided in continuous manner to ensure its sustained efficacy. Our study demonstrates that consumption of zinc fortified rice as a staple food can be efficient, even with a relative small amount of zinc as fortificant.

Hess et al. [38] suggested that co-fortification of zinc with other micronutrients could affect zinc absorption. However, there is a lack of literature on interactions between zinc and other micronutrients when used combined as fortificants on zinc absorption except on folic acid [43], which does not appear to effect zinc absorption. The results from the present study support the latter, because we found a positive impact on serum zinc concentration even though the rice that was co-fortified with, iron, and several vitamins, suggesting that zinc absorption is not hampered by the addition of other micronutrients. Studies in school children in NE Thailand [44] and Vietnam [45] with foods fortified with multiple micronutrient including iron and zinc also showed impacts on serum zinc concentrations, suggesting that inhibition of zinc absorption is less a problem in fortified foods than in supplements. 

### 4.2. MMFR Increased Folate Concentration and Reduced Deficiency

In the present study, MMFR with folic acid and other micronutrients significantly increased the serum folate of school children by 2.25 ng/mL compared to children in unfortified rice group. Folate concentrations increased in both groups over time. This could have been due to higher dietary intake of folic acid at endline as compared to baseline. Another possibility is that folate concentrations have deteriorated in the samples during storage at −20 °C, as folate has been shown to be very sensitive to degradation, even when stored at −20 °C for the period of more than 6 months [46]. However, the recent national micronutrient survey 2014 reported a prevalence of folate deficiency of 7.9% and 17.8% in children and women, respectively [7], which was similar to our baseline prevalence. Regardless of the underlying cause of the change in folate concentrations from baseline to endline, the present study showed an increase in folate concentrations similar to other studies. Studies on fortification with the same amount of folic acid in our study (0.14 mg/100 g) in flour and grain products in the United States and Canada showed almost doubling of serum folate concentrations (4.6–10 ng/mL) [47,48]. Another study on fortification with low amounts of folic acid in young women also had a significant effect on folate status [49], supporting our findings and suggesting that fortified rice could be an effective means to increase folate status among women of reproductive age, and thereby can contribute to the prevention of neural tube defects. 

Previous studies brought up issue of the appropriate level of folic acid for fortification in staple food, e.g., wheat, maize, and rice [47,50,51]. It was suggested that the amount of folic acid for fortification should account for bio-availability for absorption and cooking method as well as the target population and expected health benefit. It was shown that different cooking techniques reduced the amount of folic acid in fortified rice by only 15–35% [52], making rice a suitable vehicle for fortification with folic acid. 

The study had a number of strengths and limitations. First, it was one of the few studies investigating the effects of fortified food on micronutrient status of school age children. Second, the study was a clustered randomized-placebo-control trial, with the largest sample size to date. A cluster-randomization was chosen because the schools had one kitchen each, and separate preparations of school meals were not feasible. The placebo-control trial allowed a distinct comparison of the different intervention groups with fortified rice to placebo (normal rice). Double blinded trial design also added strength to the study. However, a limitation in this study was that the FORISCA study was meant for effectiveness rather than efficacy; therefore, control on the cooking and preparation process and dietary intake of participating schoolchildren were not conducted. Additionally, participation of children at follow up was not enforced, which resulted in a reduction of the number of samples at midline and endline. Most of the absences at follow up were due to reasons other than drop-out. Finally, due to limited funding, only one type of fortified rice (NutriRice) was selected to be studied on effect of MMFR on folate status as compared to normal rice, assuming that the impact of folic acid fortification in one fortified rice group would represent the effect of the other fortified rice groups as they were produced using similar extrusion methods. A cluster-randomization was chosen because the schools had one kitchen each, and separate preparations of school meals were not feasible.

## 5. Conclusions

In summary, we have shown an alarming high prevalence of zinc deficiency, based on serum zinc concentrations, which could be improved through the provision of rice fortified with zinc, while folate status was also improved. Fortified rice in particular can fill the gap of micronutrient intake from dietary sources during season of low dietary diversity. As rice is the staple diet for Cambodia, it is strongly suggested that MMFR should be considered to be included in the school meal program and possibilities should be explored to introduce MMFR to the general population as well.

## Figures and Tables

**Figure 1 nutrients-11-02843-f001:**
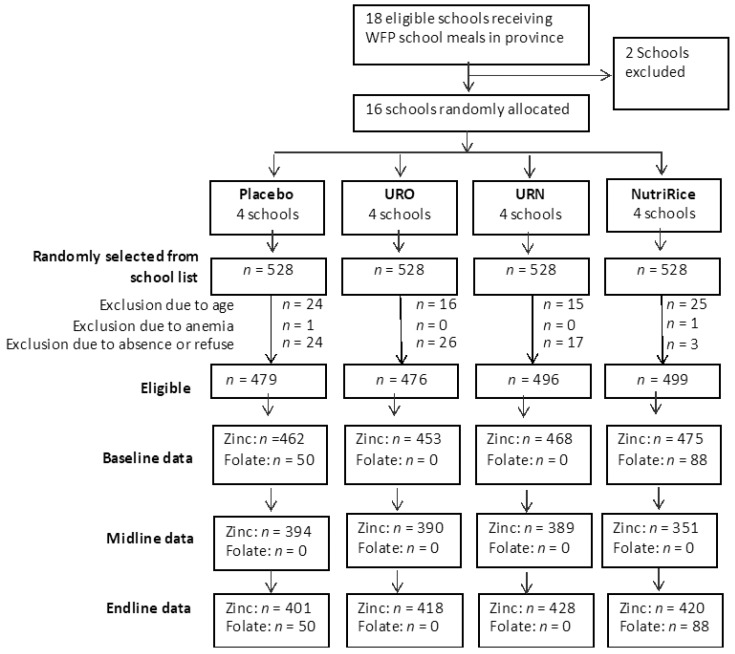
Flow chart of the study.

**Table 1 nutrients-11-02843-t001:** Micronutrient contents of normal rice, and the actual and target values of the different types of fortified rice, per 100g of uncooked blended rice.

Micronutrients	Normal Rice ^1^	URO	URN	NutriRice	Target Value
Iron (mg)	0.3	10.67	7.55	7.46	7.26
Zinc (mg)	1.0	3.04	2.02	3.68	3.50
Vitamin B1 (mg)	ND	1.06	1.43	0.69	0.60
Folic Acid (mg)	ND	0.17	0.28	0.14	0.20
Vitamin A (mg)	0.03	-	0.64	0.29	0.30
Vitamin B3 (mg)	ND	-	12.57	7.98	8.00
Vitamin B12 (μg)	0.4	-	3.8	1.3	1.2
Vitamin B6 (mg)	0.1	-	-	0.92	0.65

Abbreviations: URO: UltraRice original formula, URN: UltraRice new formula, ND: no data. ^1^ Composition of normal rice based on the analyses of 10 rice samples.

**Table 2 nutrients-11-02843-t002:** Baseline characteristics of children in the study.

Outcomes	Placebo	URO	URN	NutriRice
*n*	462	453	468	475
Age (y)	9.62 ± 2.29	9.63 ± 2.18	9.61 ± 2.20	9.73 ± 2.42
% girls	50.2 (*n* = 232)	49.4 (*n* = 224)	50.0 (*n* = 234)	49.9 (*n* = 237)
% inflammations	42.9 (*n* = 195)	42.4 (*n* = 191)	32.5 (*n* = 151)	32.2 (*n* = 150)
% parasite infection	22.9 (*n* = 78)	25.8 (*n* = 90)	19.8 (*n* = 70)	12.9 (*n* = 39)
% Hemoglobinopathy (Hb E ≥5%)	41.8 (*n* = 193)	43.9 (*n* = 199)	43.4 (*n* = 203)	41.7 (*n* = 198)
**Anthropometric status**				
Underweight (WAZ ≤2) (only for <10 years)	43.1% (*n* = 109)	40.1% (*n* = 108)	49.6% (*n* = 135)	40.1% (*n* = 107)
Stunted (HAZ ≤2)	43.8% (*n* = 202)	39.6% (*n* = 179)	44.5% (*n* = 208)	45.5% (*n* = 214)
Wasted (BAZ ≤2)	25.4% (*n* = 117)	27.0% (*n* = 122)	29.8% (*n* = 139)	21.5% (*n* = 101)
**Micronutrient status**				
Hb (g/L)	123.5 ± 10.2	124.2 ± 9.8	123.7 ± 9.2	123.9 ± 10.4
% anemia	18.7 (*n* = 86)	15.3 (*n* = 69)	17.5 (*n* = 82)	17.2 (*n* = 81)
ID total	55.2 (*n* = 251)	56.8 (*n* = 256)	48.6 (*n* = 226)	49.0 (*n* = 229)
ID with anemia	11.9 (*n* = 54)	10.2 (*n* = 46)	10.5 (*n* = 49)	10.3 (*n* = 48)
Body Iron (mg/kg)	6.49 ± 2.0	6.53 ± 2.2	6.16 ± 2.4	6.16 ± 2.4
% 0 < BI < 4 mg/kg	9.0 (*n* = 41)	10.9 (*n* = 49)	11.6 (*n* = 54)	13.3 (*n* = 62)
% BI < 0 mg/kg	1.3 (*n* = 6)	1.3 (*n* = 6)	3.0 (*n* = 14)	2.6 (*n* = 12)
RBP (µmol/L)	1.62 ± 0.43	1.68 ± 0.43	1.48 ± 0.43	1.51 ± 0.43
% marginal VA status	7.5 (*n* = 34)	3.8 (*n* = 17)	13.8 (*n* = 64)	12.4 (*n* = 58)
% VAD	0.7 (*n* = 3)	0.2 (*n* = 1)	1.3 (*n* = 6)	1.1 (*n* = 5)
Serum Zinc (µmol/L)	7.7 ± 1.86	7.74 ± 1.69	7.61 ± 1.70	8.11 ± 1.88
% zinc deficiency	89.4 (*n* = 413)	90.1 (*n* = 408)	92.3 (*n* = 432)	83.6 (*n* = 397)
% Severe zinc deficiency (<7.6 µmol/L)	49.8 (*n* = 230)	49.7 (*n* = 225)	53.8 (*n* = 252)	43.6 (*n* = 207)
Serum Folate (ng/mL)	5.40 ± 1.60	N/A	N/A	6.25 ± 2.06
% Folate deficiency (<4 ng/mL)	18.0 (*n* = 9 out of 50)	N/A	N/A	9.1 (*n* = 8 out of 88)

Results are mean ± SD unless stated, WAZ: Weight-for-age-z-scores, HAZ: height-for-age z-scores, BAZ: BMI-for-age-z-scores, Hb: hemoglobin, ID: Iron deficiency, BI: body iron, RBP: Retinol Binding Protein, VA: Vitamin A, VAD: Vitamin A deficiency, N/A: not available, URO: UltraRice original formula, URN: UltraRice new formula.

**Table 3 nutrients-11-02843-t003:** Dietary information and attendance of each intervention group.

Variables	Placebo	URO	URN	NutriRice
**Attendance**				
*n*	462	453	468	475
Number of school days over the whole study period	127	126	126	132
% of attendance over the study period	86.6%	82.0%	83.6%	84.1%
Number school days in the BL–ML period	75	81	82	86
% attendance over the BL–ML period	88.5%	85.0%	87.7%	87.2%
Number of school days in the ML–EL period	68	61	62	52
% attendance over the ML–EL period	87.6%	88.4%	87.7%	86.5%
% attendance within 10 days before ML	89.9%	87.7%	88.4%	87.0%
% attendance within 10 days before EL	86.3%	86.5%	87.8%	85.7%
**Zinc intake**				
Zinc in fortified rice, mg/100g	-	3.0	2.0	3.7
Estimated zinc intake at school meal (%RDA)*	-	3.5 (44%)	2.3 (29%)	4.2 (53%)
Estimated zinc intake per child from BL–EL, mg	-	441.0	292.3	558.4
Estimated zinc intake per child from BL–ML, mg	-	283.5	190.2	363.8
Estimated zinc intake per child from ML–EL, mg	-	213.5	143.8	219.9
**Molar ratio**				
Phytate content, mg/100g of cooked rice	108.9	108.9	108.9	108.9
Phytate:zinc molar ratio	-	3.55	5.34	2.93
**Intake of folic Acid**				
N	50			88
Folic acid in fortified rice, mg/100g	-	0.17	0.28	0.14
Estimated folic acid intake per meal (% RDA)*	-	0.2 (50%)	0.32 (80%)	0.16 (40%)
Estimated folic acid intake per child from BL-EL, mg	-	25.4	35.3	21.1

* meal ratio corresponding to 115 g uncooked rice. BL: Baseline, ML: Midline, EL: Endline, RDA: Recommended daily allowance, URO: UltraRice original formula, URN: UltraRice new formula.

**Table 4 nutrients-11-02843-t004:** Biochemical outcomes of serum zinc concentration of participating children after three and six months of intervention.

Time Point	Group	*n*	Mean (μmol/L)	SE	Estimated Effect Coefficient ^1^ (95% CI)	*p*-Value
Baseline	Placebo	462	7.83	0.08	-	
	URO	453	7.83	0.08	-	
	URN	468	7.67	0.08	-	
	NutriRice	475	8.15	0.08	-	
Midline	Placebo	393	9.71	0.09	-	
	URO	389	8.69	0.09	−1.02 (−1.43; −0.61)	<0.001
	URN	389	9.72	0.09	−0.17 (−0.24; 0.58)	0.422
	NutriRice	351	9.54	1.00	−0.50 (−0.93; 0.07)	0.024
Endline	Placebo	401	7.57	0.09	-	
	URO	418	8.56	0.09	0.98 (0.58; 1.38)	<0.001
	URN	428	8.27	0.09	0.85 (0.45; 1.25)	<0.001
	NutriRice	420	9.29	0.09	1.40 (0.98; 1.82)	<0.001

^1^ Interaction term of Group x time (GLM mixed model adjusted for age, gender, hemoglobinopathy, body iron. Hb concentration, anaemia parasiste infection, and inflammation); URO: UltraRice original, URN: UltraRice new, SE: Standard Error of the mean, CI: confidence interval.

**Table 5 nutrients-11-02843-t005:** Prevalence of zinc deficiency of participating children after three and six months of interventic.

				Zinc Deficiency			Severe Zinc Deficiency	
				(<9.9 µmol/L)			(<7.6 µmol/L)	
Time Point Group		*n*		Estimated Effect			Estimated Effect	
			%	Adjusted OR^1^	*p*-Value	%	Adjusted OR^1^	*p*-Value
				(95% CI)			(95% CI)	
	Placebo	462	89.4	-		49.8	-	
	URO	453	90.1	-		49.7	-	
Baseline	URN	468	92.3	-		53.8	-	
	NutriRice	475	83.6	-		43.6	-	
	Placebo	393	60.6	-		15.5	-	
	URO	389	79.2	2.25(1.23;4.10)	0.08	22.4	1.56(0.93;2.62)	0.09
Midline	URN	389	55.8	0.05(0.30;1.02)	0.059	15.4	0.79(0.46;1.37)	0.4
	NutriRice	351	60.1	1.25(0.69;2.26)	0.461	17.7	1.36(0.76;2.42)	0.29
	Placebo	401	93.3	-		53.6	-	
	URO	418	80.6	0.25(0.12;0.52)	<0.001	29.7	0.35(0.22;0.55)	<0.001
Endline	URN	428	83.4	0.25(0.11;0.53)	<0.001	37.1	0.39(0.25;0.60)	<0.001
	NutriRice	420	65.7	0.16(0.08;0.34)	<0.001	25.7	0.28(0.17;0.47)	<0.001

^1^ Interaction term of Group x time (GLM mixed model adjusted for age, gender, hemoglobinapathy, bodyiron, Hb concentration, anaemia, parasiste infection and inflammation.

**Table 6 nutrients-11-02843-t006:** Biochemical outcomes of folate concentration and deficiency of participating children.

Biochemical Indicator	Placebo	Nutririce
Serum Concentration		
*n*	50	88
Baseline (± SD), ng/mL	5.40 ± 1.60	6.25 ± 2.06
Endline (± SD), ng/mL	7.13 ± 1.70	9.54 ± 2.76
Coefficient ^1^ (95% CI)	-	2.25 * (0.63; 3.87)
Deficiency Prevalence		
*n*	50	88
Baseline (%)	18.0%	9.1%
Endline (%)	2.0%	0%
Adjusted OR ^1^ (95% CI)	-	1.59 (0.11; 23.47)

^1^ Interaction term of Group x time (GLM mixed model adjusted for age, gender, hemoglobinopathy, body iron, Hb concentration, anaemia, parasiste infection, and inflammation). * Siginificant difference of group at endline compared to baseline. SD: Standard Deviation, OR: Odds Ratio, CI: Confidence interval.

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
