# Peer review of "Multi-Micronutrient Fortified Rice Improved Serum Zinc and Folate Concentrations of Cambodian School Children. A Double-Blinded Cluster-Randomized Controlled Trial"

_nutrients, 2019, doi:10.3390/nu11122843_

Round 1

Reviewer 1 Report

General comments

The manuscript of Kuong et al. investigates the effects of rice micronutrient fortification on the zinc and folate status of Cambodian school children. At the target level of 3.5 mg zinc/100 g of uncooked blended rice, the data presented in the manuscript support the notion that zinc fortification of rice raises serum zinc levels and decreases the prevalence of zinc deficiency during the lean season (April-July) in Cambodia. Moreover, six months of folate-fortified rice consumption raised serum folate levels somewhat more than placebo normal rice. The effect of folate-fortified rice on folate deficiency was not significant at the end of the study because overall the prevalence of folate deficiency was low in this population.

Specific comments

1) Line 58-59, “Micronutrient deficiencies among preschool children was also highly prevalent, with 67.5% and 20.2% of them being classified as deficient in zinc and folate respectively (8).” Please specify in this sentence the blood zinc and folate concentrations under which subjects are classified as deficient even though the information is mentioned later in the text.

2) Table 1. Please provide the micronutrient composition of the “normal” rice in the placebo group.

3) Tables 2 and 3 are the same. The unit in which serum zinc concentrations are tabled should be specified in the column. Please include blood folate concentrations (and unit) at baseline even though the information is mentioned in Table 6. Specify the folate concentration threshold for deficient. Indicate the sample size (n) for % folate deficiency occurrence, for instance Placebo: 18.0% (9 subjects out of 50), NutriRice: 9.1% (8 subjects out of 88).

4) Table 4. It is unusual and unexpected that SE is the same (1.24) regardless of the group and time point. Please double check. The unit in which serum zinc concentrations are tabled should be specified either in the table caption or in the column header.

5) Table 6. The meaning of the superscripts (a, b) was not provided.

6) Line 278. Please explain how a zinc intake of 3-5 mg/day in the present study was reached. It would help to explain as quantitatively as possible how the authors accounted for the zinc these children consumed in addition to the zinc provided by the school meal program. Same comment applies to folate intake.

7) Lines 279-281. Aspects of this sentence are redundant.

8) The discussion section would be strengthened by expanding on (1) possible refinements to rice fortification to lower further the prevalence of zinc deficiency in this population, and (2) possible alternatives to rice fortification such as accessibility to preserved foods during the lean season.

Author Response

Reviewer 1

General comments

The manuscript of Kuong et al. investigates the effects of rice micronutrient fortification on the zinc and folate status of Cambodian school children. At the target level of 3.5 mg zinc/100 g of uncooked blended rice, the data presented in the manuscript support the notion that zinc fortification of rice raises serum zinc levels and decreases the prevalence of zinc deficiency during the lean season (April-July) in Cambodia. Moreover, six months of folate-fortified rice consumption raised serum folate levels somewhat more than placebo normal rice. The effect of folate-fortified rice on folate deficiency was not significant at the end of the study because overall the prevalence of folate deficiency was low in this population.

Specific comments

1) Line 58-59, “Micronutrient deficiencies among preschool children was also highly prevalent, with 67.5% and 20.2% of them being classified as deficient in zinc and folate respectively (8).” Please specify in this sentence the blood zinc and folate concentrations under which subjects are classified as deficient even though the information is mentioned later in the text.

Author reply.

We have added the values defined as cut-off.

2) Table 1. Please provide the micronutrient composition of the “normal” rice in the placebo group.

Author Reply.

We’ve added the results of an analysis of 10 normal rice samples as ‘normal rice’ to the table.

3) Tables 2 and 3 are the same.

Author Reply.
Sorry for this omission. The correct Table 3 has been added.

The unit in which serum zinc concentrations are tabled should be specified in the column.

Author reply

Done.

Please include blood folate concentrations (and unit) at baseline even though the information is mentioned in Table 6. Specify the folate concentration threshold for deficient.

Author reply

Done

Indicate the sample size (n) for % folate deficiency occurrence, for instance Placebo: 18.0% (9 subjects out of 50), NutriRice: 9.1% (8 subjects out of 88).

Author reply

Done.

4) Table 4. It is unusual and unexpected that SE is the same (1.24) regardless of the group and time point. Please double check. The unit in which serum zinc concentrations are tabled should be specified either in the table caption or in the column header.

Author reply. Thank you for pointing out the SEs. Actually, the SEs are almost the same between the groups, but lower than reported in the first table. They have been corrected now. And we have added the unit for zinc in the column head.

5) Table 6. The meaning of the superscripts (a, b) was not provided.

Author reply.

Superscripts have been removed.

6) Line 278. Please explain how a zinc intake of 3-5 mg/day in the present study was reached.

It would help to explain as quantitatively as possible how the authors accounted for the zinc these children consumed in addition to the zinc provided by the school meal program.

Author reply. This is explained in lines 260 – 262 : “Zinc intake per each group for each child ranged from 2.3 to 4.2 mg per meal (day) which contribute about 29-53% of zinc RDA (of 8 mg/d).” We have changed the 3-5mg/day to 2.3-4.2 mg/d to be more specific.

Same comment applies to folate intake.

Author reply : lines 264 – 266 : “. The intake of folic acid per meal (day) was highest in URN (0.32mg) compared to NutriRice (0.16mg) and URO (0.2mg).”

7) Lines 279-281. Aspects of this sentence are redundant.

Author reply. Sentence has been improved

8) The discussion section would be strengthened by expanding on (1) possible refinements to rice fortification to lower further the prevalence of zinc deficiency in this population, and (2) possible alternatives to rice fortification such as accessibility to preserved foods during the lean season.

Author reply

We have added a sentence to the discussion on this (lines 345-347)

Reviewer 2 Report

This is an exceptionally well written manuscript describing a well designed and executed study examining the impact of multiple micronutrient fortification of rice on the zinc and folate status in school children in Cambodia.

I only have two minor comments for clarification.

Table 1 shows the micronutrient content of uncooked rice.  How does this change once the rice is cooked?

Line 185 refers to an amount of 115 g consumed per day.  Is this uncooked or cooked rice?

Author Response

Reviewer 2

This is an exceptionally well written manuscript describing a well designed and executed study examining the impact of multiple micronutrient fortification of rice on the zinc and folate status in school children in Cambodia.

I only have two minor comments for clarification.

Table 1 shows the micronutrient content of uncooked rice.  How does this change once the rice is cooked?

Author reply. In another paper, we have shown that zinc (and iron) is (are) stable during cooking, hence no change in overall amount. Of course, because of the addition of water, the zinc concentration diminishes.

Line 185 refers to an amount of 115 g consumed per day.  Is this uncooked or cooked rice?

Author reply: This is uncooked rice. After cooking, this becomes about 400 – 450 g of wet rice.